# Effects of Afforestation on Soil Bulk Density and pH in the Loess Plateau, China

**Xiaofang Zhang [1], Jan F Adamowski [2], Ravinesh C Deo [3] 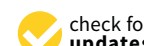, Xueyun Xu [1], Guofeng Zhu [1] and Jianjun Cao [1,\*]**

[1]  College of Geography and Environmental Science, Northwest Normal University, Lanzhou 730070, China; 15726487062@163.com (X.Z.); xuxueyun@nwnu.edu.cn (X.X.); gfzhu@lzb.ac.cn (G.Z.)
[2]  Department of Bioresource Engineering, Faculty of Agricultural and Environmental Sciences, McGill University, Sainte Anne de Bellevue, QC H9X 3V9, Canada; jan.adamowski@mcgill.ca
[3]  School of Agricultural, Computational and Environmental Sciences, Centre for Sustainable Agricultural Systems & Applied Climate Sciences, University of Southern Queensland, Springfield Central, QLD 4300, Australia; ravinesh.deo@usq.edu.au
\*  Correspondence: caojj@nwnu.edu.cn; Tel.: +86-0931-7971565

**Abstract:** Sustainable land management requires a clear understanding of the changes in soil quality. In exploring whether afforestation has the potential to improve the soil quality in China's Loess Plateau, soil bulk density ($\rho_s$) and pH were compared under five treatments: three forested treatments (16-and 40-year-old apricot stands, and 40-year-old poplar stands), and individual abandoned and cultivated treatments, serving as the controls. Bulk density across the 0–1.0 m soil profile under the 16-year-old apricot treatment (1.12 Mg m$^{-3}$) and 40-year-old poplar treatment (1.16 Mg m$^{-3}$) were significantly smaller than their counterparts under the cultivated (1.20 Mg m$^{-3}$) and abandoned treatments (1.23 Mg m$^{-3}$). Soil pH of the cultivated treatment (8.46) was significantly lower than that of the abandoned treatment (8.51) or than that of any forested treatment. The $\rho_s$ and pH were both affected by stand age, with the $\rho_s$ and pH of the 40-year-old apricot treatment being 0.10 Mg m$^{-3}$ and 0.05 units greater, respectively, than those of the 16-year-old apricot treatment. Treatment and soil depth appeared to interact to influence the $\rho_s$, but this same interaction did not influence the soil pH. This study suggested that afforestation species and stand age should be taken into consideration to harvest maximum benefits from the afforestation efforts.

**Keywords:** soil erosion; soil quality; stand type; stand age; tree plantation

## 1. Introduction

Afforestation is often regarded as a strategy to reduce soil erosion, assist in nutrient cycling [1], alter local hydrology [2], and to sequester carbon [3]. Many large-scale afforestation programs have been implemented around the world since 1990, and such planted forests are projected to cover up to $3.0 \times 10^6$ km$^2$ by 2020. In China, significant environmental problems such as soil erosion [4], extensive desertification [5,6], declining land productivity [6], and land salinization [7] prompted the government to adopt afforestation to address these issues [5]. The focus was on lands suffering from poor soil fertility and productivity [8]. Through the implementation of a series of protective projects (e.g., Three Norths Shelter Forest System Project, Natural Forest Conservation Program, Sand Control Program, and Grain for Green Project), the Chinese government has planned to expand forests by $0.44 \times 10^6$ km$^2$ between 2006 and 2030 [9], and to increase forest cover from 22% in 2018 to 26% by 2050 [10].

Serious soil erosion and soil losses on China's Loess Plateau in recent decades have threatened the region's sustainable development [11]. Accordingly, the Loess Plateau became a key area for

afforestation [12]. Many studies found afforestation to have a positive impact, such as decreasing soil bulk density ($\rho_s$) and reducing the pH of alkaline soils [13]. However, given the current lack of ground-truth observations, the regional impact of afforestation remains poorly understood [14]. The limited availability of information on the effect of afforestation on $\rho_s$ and pH has made it difficult for regional policy makers to develop sustainable socioeconomic and ecological development plans [15].

The $\rho_s$ refers to the mass of an oven-dry sample of undisturbed soil per unit bulk volume [16]. It is a key indicator of soil quality, and plays a fundamental role in determining a soil's physicochemical state, nutrient pool [17], and organic carbon storage [18], along with its ability to sustain plant growth [19]. Compared to denser soils, soils of lesser density (lower $\rho_s$) generally exhibit better soil structure and provide a greater ability to retain water, nutrients, and organic carbon [18]. Moreover, a lower $\rho_s$ is also helpful in regulating the movement of fluids and gases within a soil, and between a soil and its interface with the environment [18,20].

Soil pH, affecting both chemical and biological reactions, serves as an important indicator of a soil's acidity or alkalinity, and plays a significant role in determining a soil's physicochemical and biological status. It also regulates critical functions such as the availability of essential nutrients, the solubility of non-essential and/or toxic elements, cation exchange capacity, and microbial activity [21]. Therefore, the distribution of soil pH provides basic and useful information that is relevant to soil management and agricultural production [21,22].

Both $\rho_s$ and pH are very sensitive to changes in land cover [22] and to the changes in land use type [23]. For example, Jiao et al. [24] found that shrub land soils in the Loess Plateau's Yanhe watershed exhibited a greater $\rho_s$ than did the region's other soils. Likewise, Templer et al. [25] reported lower soil pH and base cations in agricultural vs. regenerating forest sites in the Dominican Republic. Following a chronosequence in afforested areas in Vestskoven, Denmark, Ritter et al. [26] noted that afforestation slowly modified the soil properties of former cultivated soils. After 30 years, a pH decrease from 6 to 4 in the upper 50 mm of soil was the most apparent change. Wang et al. [13] indicated that after returning farmland to forests, the $\rho_s$ tended to decrease. These studies have therefore motivated the current study, which aims to explore the changes in $\rho_s$ and pH in the soils of China's Loess Plateau after afforestation. Afforestation in this study is hypothesized to decrease in $\rho_s$ and pH over time, as stand age increases.

## 2. Materials and Methods

### 2.1. Study Area

Located in central Gansu Province, China, Huining County (104°29′–105°31′ E, 35°24′–36°26′ N), the present study area is located in the south-to-north sloping Loess Plateau. With an average altitude of 2025 m and an area of roughly 6439 km$^2$, the region is situated in a temperate continental climate zone with a mean annual temperature ranging from 6 °C to 9 °C, and a mean rainfall of 180 to 450 mm y$^{-1}$, of which 58% falls between July and September. The mean evaporation rate in Huining County is over 1800 mm y$^{-1}$. The region is characterized by complex tectonic structures, most of which are based on metamorphic rocks and granites. The Quaternary deposits (loess) are widely distributed as landform types of low-hill, loess-hill, and loess hill-ridge, accounting for more than 95% of the total area [27]. The soil particles are dominated by silt (above 60%), and clay and sand accounts for 5% and 30%, respectively [28]. The field capacity is between 13% and 25%, and the permanent wilting point is between 3% and 8% [15]. Most of these soils have been significantly disturbed by humans in the past, when these lands were shaped by forming terraces for agricultural production in the 1970s. This could lead to the differences in both $\rho_s$ and pH that are significant but small throughout the 1 m profile.

In 1999, implementing the national policy of returning farmlands to forest, Huining County began extensive afforestation. By the end of 2015, the afforested area in Huining had reached 707 km$^2$, and 12.47% of the county was forest-covered. Due to water deficiency issues, tree survival rates were

very low in the county's northern and central regions, but higher in the south where precipitation was greater [15]; accordingly, southern Huining was selected as the study area.

## 2.2. Experimental Design

In exploring the effects of afforestation on $\rho_s$ and pH, five different treatments, including three forested treatments (16- and 40-year-old apricot stands and 40-year-old poplar stands), one abandoned treatment, and one cultivated treatment, were studied over the period of July to September 2017 (Table 1). Cultivated and abandoned treatments in the study area were investigated as a basis of comparison for the forested treatments. The crops in southern Huining were dominated by wheat (*Triticum æstivum* L.), maize (*Zea mays* L.), potato (*Solanum tuberosum* L.) and flax (*Linum usitatissimum* L.) [15]. Afforestation species were mainly poplar (*Populus tremula* L.) and apricot (*Prunus armeniaca* L.). Most poplar stands were 40 years old (40-year-old poplar treatment), having been planted in 1978 as part of the Three Norths Shelter Forest System Project. The apricot stands were either planted in 1978 (40-year-old apricot treatment) or in 2001 (16-year-old apricot treatment).

**Table 1.** The number of plots in each of the five treatments and soil bulk density ($\rho_s$) and pH (to 1.0 m soil depth) of the different treatments in each of the three blocks. (mean ± standard deviation).

| Variables | $\rho_s$ | | | | Soil pH | | | |
|---|---|---|---|---|---|---|---|---|
| Treatments | Block 1 | Block 2 | Block 3 | df | Block 1 | Block 2 | Block 3 | df |
| Apricot (16 years) (27 plots) | 1.11 ± 0.08A | 1.11 ± 0.08A | 1.16 ± 0.07B | 2 | 8.41 ± 0.09A | 8.52 ± 0.14B | 8.53 ± 0.15B | 2 |
| Apricot (40 years) (27 plots) | 1.19 ± 0.06A | 1.24 ± 0.091B | 1.22 ± 0.08C | 2 | 8.68 ± 0.22A | 8.52 ± 0.15B | 8.49 ± 0.11B | 2 |
| Poplar (40 years) (27 plots) | 1.10 ± 0.06A | 1.09 ± 0.07A | 1.28 ± 0.08B | 2 | 8.46 ± 0.64A | 8.44 ± 0.13A | 8.60 ± 0.13B | 2 |
| Cultivated treatment (91 plots with 4 kinds of crops lumped) | 1.15 ± 0.09A | 1.28 ± 0.10B | 1.34 ± 0.12C | 2 | 8.44 ± 0.16A | 8.49 ± 0.38B | 8.38 ± 0.15C | 2 |
| Abandoned treatment (> 5 years, 52 plots) | 1.15 ± 0.11A | 1.27 ± 0.08B | 1.31 ± 0.07C | 2 | 8.49 ± 0.15A | 8.55 ± 0.16B | 8.60 ± 0.13C | 2 |

**Notes:** Different uppercase letters show differences between different blocks for a given treatment; df = degree of freedom.

Industrialization and urbanization have led to the abandonment of cultivated lands of varying yield potential, driving migrant workers to leave the land for urban jobs, and inciting families to move to Huining and other cities for better educational opportunities. This, in turn, has led to few young people returning after college graduation. Most of the abandoned tracts of land in the study area had been abandoned for between 5 and 10 years, while a few had been abandoned for over 20 years [15]. Given the difficulty in identifying the year in which a farm was definitively deserted, in the present study, abandoned lands were classified as those having been so for over five years.

In total, three blocks, each housing all five different treatments, were selected (Figure 1). Blocking was done to reduce the unexplained variability (i.e., to reduce the error). Differences in $\rho_s$ and pH under each treatment among the three blocks are presented in Table 1. One should note that it is a good sign that the blocks appear to have an influence on the results, as this variability can be removed from the unexplained variability. In forested treatments, soils were sampled from nine 10 m × 10 m plots that were arranged at a 10 m spacing both up–down and on the contour in the middle slope, while in the abandoned and the cultivated treatments, soils were sampled from three or five 10 m × 10 m plots arranged at a 10 m spacing along the horizontal contour, due to the limited areas of these lands. In a small block, if the replicated plots could represent the whole block, this non-random sampling method was also feasible, as has been advocated by several other researchers (e.g., [15,23,29,30]). To improve the comparability between the forested treatments and the non-forested treatments, a majority of the non-forested lands were sampled in the vicinity of

forested lands. In each plot, three 1 m × 1 m sub-plots, spaced along the diagonal (both ends and midpoint), were excavated to a soil profile depth of 1.0 m, a depth representing the extent of vertical root distribution of herbaceous and woody plants in this region [9,31]. In the poplar treatment, the physical barrier of litterfall (Figure 2B) contributed to a lack of vegetative cover and low plant diversity [32], and the above-ground biomass (AGB) was not collected. In the apricot treatment and abandoned treatment, the living AGB was collected by the harvest method, following earlier studies [33]. In the cultivated treatment, there was a lack of living AGB due to the growth of crops and regular weeding.

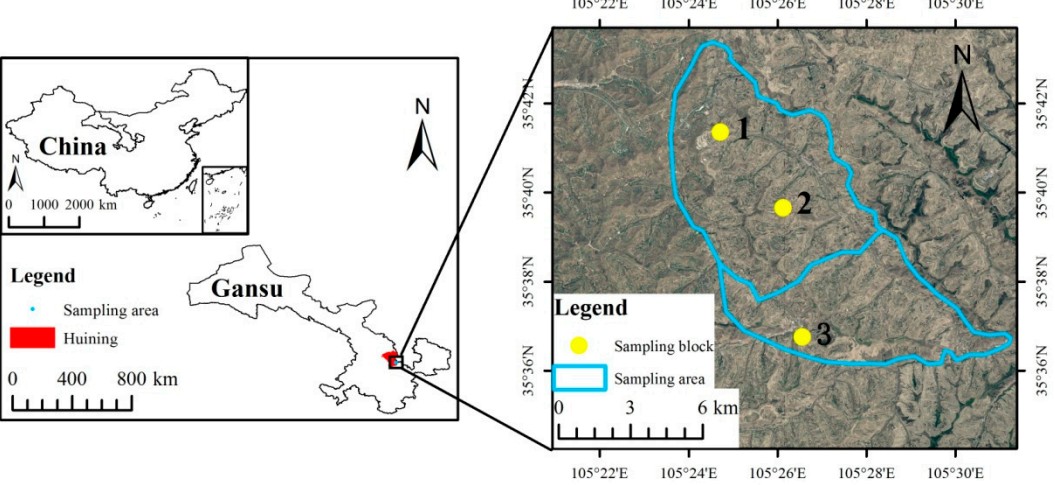

**Figure 1.** Location of the three experimental blocks in Huining County, Gansu Province, China.

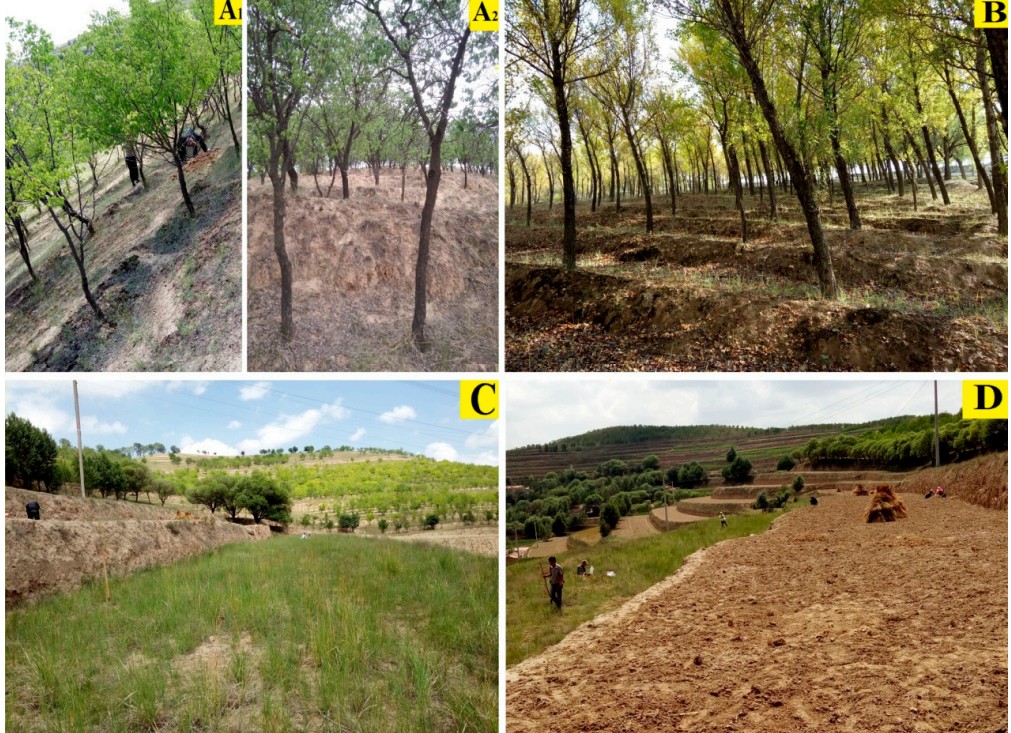

**Figure 2.** Sampled treatments in the study. **A₁**: 16-year-old apricot stand; **A₂**: 40-year-old apricot stand; **B**: 40-year-old poplar stand; **C**: Abandoned treatment; **D**: Cultivated treatment.

In each sub-plot, soil samples were taken at depths of 0–0.1 m, 0.1–0.2 m, 0.2–0.4 m, 0.4–0.6 m, 0.6–0.8 m, and 0.8–1.0 m, using a cutting ring (volume, $1 \times 10^{-4}$ m³). Compared to other methods,

this approach provided us a better comparison of soil properties at multiple depths [29] and has, accordingly, been used widely (e.g., [34,35]). Furthermore, the topsoil and the deeper soil profiles may exhibit different responses of $\rho_s$ and pH as they do for soil moisture [15].

The number of plots in each treatment is presented in Table 1, while the stand density (SD) and AGB are presented in Table 2.

**Table 2.** Stand density (SD) and above-ground biomass (AGB) of the 16- and 40-year-old apricot treatments, the poplar treatment, and abandoned treatment (mean ± standard deviation).

| Variables | Apricot (16 years) | Apricot (40 years) | Poplar (40 years) | Abandoned Treatment |
|---|---|---|---|---|
| SD (tree ha$^{-1}$) | $1030 \pm 287$A | $778 \pm 352$B | $2252 \pm 856$ | — |
| AGB (g) | $24.94 \pm 12.35$A | $18.98 \pm 7.53$B | — | $43.41 \pm 22.16$ |

Notes: '—' represents a lack of samples for the particular treatment. Different uppercase letters indicate a significant difference among different treatments. ($p \leq 0.05$).

### 2.3. Plant and Soil Analysis

Each soil sample collected with a cutting ring was stored in a Ziplock$^{TM}$ bag and weighed on site. Upon return to the lab, the soil sample in each bag was transferred to an aluminum box of known weight. Samples were dried at 105 °C for 10 hr, reweighed, and $\rho_s$ was calculated as:

$$\rho_s = \frac{\text{dry weight}}{\text{volume}}$$

Soil pH of a 2:5 soil: water suspension was measured using a Sartorius PB-10 pH meter. Plant samples were oven-dried at 80 °C to a constant weight (48 hr), and biomass was expressed as the dry weight (g).

### 2.4. Statistical Analysis

The data were analyzed using SPSS 22.0 (SPSS Inc., Chicago, IL, USA) statistical software. All data were expressed as the mean ± standard deviation. The effect of soil depth on $\rho_s$ and pH, and the differences in measured variables among soil layers and treatments, and the difference in $\rho_s$ and pH of each treatment among three blocks, were analyzed by using a one-way analysis of variance (ANOVA). The $\rho_s$ and pH were analyzed by two-way ANOVA with treatment and soil depth as fixed factors. The least-significant-difference (LSD) test was used to compare the means of variables when the results of ANOVA were significant at $p \leq 0.05$. The Pearson's Product Moment Correlation (r) was used to identify the statistically significant relationships between $\rho_s$ and pH. Origin Pro 9.0 software was then used to visualize the data through appropriate diagnostic plots.

## 3. Results

### 3.1. Vertical Distribution of $\rho_s$ and Soil pH under Different Treatments

Under the cultivated treatment, the $\rho_s$ first increased to a peak value in the 0.2–0.4 m soil layer, and then decreased as the soil depth increased, and $\rho_s^{0 \cdot 0.1 \text{ m}}$ and $\rho_s^{0.8 \cdot 1.0 \text{ m}}$ were the lowest of all depths. Under the abandoned treatment, $\rho_s^{0.6 \cdot 0.8 \text{ m}}$ and $\rho_s^{0.8 \cdot 1.0 \text{ m}}$ were the lowest, and $\rho_s^{0 \cdot 0.1 \text{ m}}$, $\rho_s^{0.1 \cdot 0.2 \text{ m}}$, and $\rho_s^{0.2 \cdot 0.4 \text{ m}}$ were relatively higher although there was not a significant difference. The overall trend was in agreement with those observed under the cultivated treatment, except that the peak value under the abandoned treatment occurred in the 0.1–0.2 m soil layer. Under the poplar treatment, except for a significant difference between $\rho_s^{0.1 \cdot 0.2 \text{ m}}$ and $\rho_s^{0.8 \cdot 1.0 \text{ m}}$, $\rho_s$ in other soil layers was the same with a value of 1.16 Mg m$^{-3}$. Under the 16-year-old apricot treatment, $\rho_s^{0 \cdot 0.1 \text{ m}}$, $\rho_s^{0.1 \cdot 0.2 \text{ m}}$, and $\rho_s^{0.2 \cdot 0.4 \text{ m}}$ were higher than the other layers, but no significant difference was found among them. However, from 0.4 m to 0.8 m, $\rho_s$ suddenly decreased by 0.05 Mg m$^{-3}$, and then it further dropped by 0.06 Mg m$^{-3}$ from 0.8 m to 1.0 m. Under the 40-year-old apricot treatment, $\rho_s^{0 \cdot 0.1 \text{ m}}$, $\rho_s^{0.1 \cdot 0.2 \text{ m}}$ and $\rho_s^{0.2 \cdot 0.4 \text{ m}}$ were relatively low, and

no significant difference was found among them. The $\rho_s$ showed a sudden increase of 0.04 Mg m$^{-3}$ in the 0.4–0.6 m soil layer, and it reached a maximum value in the 0.6–0.8 m soil layer, and then dropped slightly (Table 3).

**Table 3.** Soil bulk density ($\rho_s$) and pH changes in different soil layers under different treatments. (mean $\pm$ standard deviation).

| | Soil Depth | Apricot (16 Years) | Apricot (40 Years) | Poplar (40 Years) | Abandoned Treatment | Cultivated Treatment | df |
|---|---|---|---|---|---|---|---|
| $\rho_s$ (Mg m$^{-3}$) | 0–0.1 m | 1.16 ± 0.07aAC | 1.22 ± 0.08acB | 1.16 ± 0.12abA | 1.22 ± 0.09acB | 1.19 ± 0.11aC | 4 |
| | 0.1–0.2 m | 1.14 ± 0.15abA | 1.21 ± 0.08acBD | 1.18 ± 0.11aB | 1.24 ± 0.0.11aCD | 1.24 ± 0.0.13bC | 4 |
| | 0.2–0.4 m | 1.16 ± 0.07aA | 1.19 ± 0.09aB | 1.16 ± 0.09abAB | 1.22 ± 0.11aC | 1.29 ± 0.11cD | 4 |
| | 0.4–0.6 m | 1.11 ± 0.06bcA | 1.23 ± 0.10bcB | 1.16 ± 0.10abC | 1.20 ± 0.11cD | 1.26 ± 0.12bE | 4 |
| | 0.6–0.8 m | 1.11 ± 0.07cA | 1.24 ± 0.07bB | 1.16 ± 0.13abC | 1.17 ± 0.12bC | 1.22 ± 0.11dD | 4 |
| | 0.8–1.0 m | 1.05 ± 0.07dA | 1.22 ± 0.08bcB | 1.13 ± 0.11bC | 1.15 ± 0.13bC | 1.18 ± 0.12aD | 4 |
| Soil pH | 0–0.1 m | 8.46 ± 0.14aAB | 8.47 ± 0.13aAB | 8.46 ± 0.13aAB | 8.49 ± 0.0.14aA | 8.45 ± 0.17aB | 4 |
| | 0.1–0.2 m | 8.47 ± 0.13acAC | 8.52 ± 0.12abB | 8.51 ± 0.15abAB | 8.52 ± 0.15abB | 8.45 ± 0.19aC | 4 |
| | 0.2–0.4 m | 8.48 ± 0.14acAC | 8.56 ± 0.14bcB | 8.52 ± 0.14bAB | 8.52 ± 0.15abB | 8.47 ± 0.16aC | 4 |
| | 0.4–0.6 m | 8.52 ± 0.14bA | 8.58 ± 0.17cB | 8.54 ± 0.15bAB | 8.53 ± 0.14bA | 8.46 ± 0.16aC | 4 |
| | 0.6–0.8 m | 8.51 ± 0.15bcAC | 8.57 ± 0.0.20bcB | 8.55 ± 0.17bAB | 8.51 ± 0.17abA | 8.47 ± 0.17aC | 4 |
| | 0.8–1.0 m | 8.49 ± 0.16abAC | 8.56 ± 0.24bcB | 8.53 ± 0.16bAB | 8.52 ± 0.16abAB | 8.46 ± 0.18aC | 4 |

**Notes:** Different uppercase and lowercase letters indicate differences at $p \leq 0.05$. Uppercase letters show differences between different treatments for a given soil depth; lowercase letters show differences across soil depth ranges within a given treatment; df = degree of freedom.

There was no difference in pH among soil layers under the cultivated treatment, whereas under the abandoned treatment, the pH in the 0–0.1 m soil layer was significantly lower than that in the 0.4-0.6 m soil layer, and values were intermediate and similar in other soil layers. Under the poplar treatment, pH in the 0–0.1 m soil layer was significantly lower than that in each soil layer below 0.2 m, but no difference between the 0.1–0.2 m soil layer, and each of the other soil layers, was found. Under the 16-year-old apricot treatment, the pH in the 0–0.1 m soil layer was significantly lower than that in the 0.4–0.6 m and 0.6–0.8 m soil layers, but no difference was found among other soil layers. In the 40-year-old apricot treatment, pH in the 0–0.1 m soil layer was significantly lower than in each soil layer below 0.2 m. Although the pH in the 0.1–0.2 m soil layer was significantly lower than that in the 0.4–0.6 m soil layer, there was no difference between it and other soil layers. Soil pH under all forested treatments first increased to a peak, and then gradually decreased with increasing soil depth. In abandoned and cultivated treatments, such trends were not apparent (Table 3).

### 3.2. $\rho_s$ and Soil pH under Different Treatments

In the 0–0.1 m soil layer, the $\rho_s$ of the 40-year-old apricot and abandoned treatments were the highest, followed by the cultivated treatment, 16-year-old apricot treatment, and poplar treatment. In the 0.1–0.2 m soil layer, $\rho_s$ of the 40-year-old apricot treatment, abandoned treatment, and cultivated treatment were higher than those of the 16-year-old apricot treatment and poplar treatment. At 0.2–0.4 m and 0.4–0.6 m, the cultivated treatment had the highest $\rho_s$ while the 16-year-old apricot treatment and poplar treatment had lower values. At 0.6–0.8 m and 0.8–1.0 m, the order of $\rho_s$ was: 40-year-old apricot treatment > cultivated treatment > abandoned treatment > poplar treatment > 16-year-old apricot treatment (Table 3). Overall, the mean $\rho_s^{0-1.0\ m}$ of the cultivated treatment and the 40-year-old apricot treatment (1.23 Mg m$^{-3}$ and 1.22 Mg m$^{-3}$, respectively) were the highest among the five treatments, while the mean $\rho_s^{0-1.0\ m}$ of the abandoned treatment was lower at 1.20 Mg m$^{-3}$ (Figure 3). In the poplar treatment, $\rho_s^{0-1.0\ m}$ was 1.16 Mg m$^{-3}$, significantly lower than that of the abandoned treatment, and significantly higher than that of the 16-year-old apricot treatment. In comparing different stand types (poplar and 40-year-old apricot), $\rho_s^{0-1.0\ m}$ was 0.06 Mg m$^{-3}$ lower in the poplar treatment than in the apricot treatment. The $\rho_s^{0-1.0\ m}$ of the apricot treatments increased by 0.10 Mg m$^{-3}$ from the 16- to the 40-year-old stands (Figure 3).

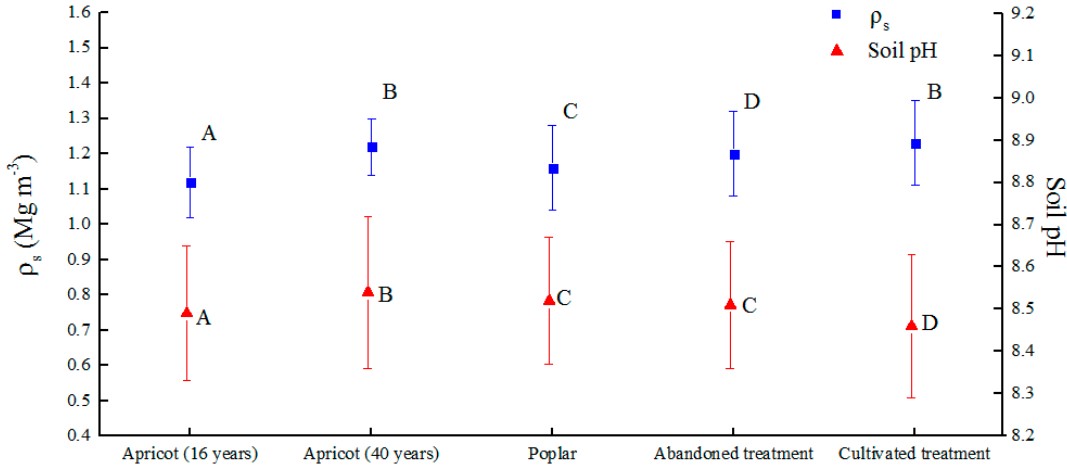

**Figure 3.** The soil bulk density ($\rho_s$) and pH across the 0–1.0 m depth soil profile under different treatments. Different uppercase letters show differences among the 16- and 40-year-old apricot treatments, the poplar treatment, abandoned treatment, and cultivated treatment at $p \leq 0.05$. Bars indicate standard deviation of the means.

In the 0–0.1 m soil layer, there was no difference in pH among the different treatments except that the pH of the cultivated treatment differed significantly from that of the abandoned treatment (Table 3). In the 0.1–0.2 m soil layer, the pH values of the 16-year-old apricot treatment and cultivated treatment were lower than those of the other treatments. In each soil layer below 0.2 m, the pH in the 40-year-old apricot treatment was high, while under the cultivated treatment it was low. In comparing the same soil layer for the two apricot treatments, pH in the 16-year-old treatment was significantly lower than that in the 40-year-old treatment, except for the 0–0.1 m soil layer. For the mean pH of the 0–1.0 m soil depth, the 40-year-old apricot treatment's pH (8.54) was significantly higher than that under other treatments (Figure 3). The abandoned treatment and the poplar treatment followed, with mean pH values of 8.51 and 8.52, respectively, with no significant difference. The mean soil pH was lowest under the cultivated treatment (8.46) and was 0.08 units lower than that under the 40-year-old apricot treatment. The mean soil pH of the 40-year-old apricot treatment was 0.02 units higher than that of the poplar treatment, and 0.05 units higher than that of the 16-year-old apricot treatment (8.49) (Figure 3).

### 3.3. Effects of Soil Depth on $\rho_s$ and pH in Different Treatments

Except for the poplar treatment, soil depth had significant effects on $\rho_s$ of other treatments, and the effect in cultivated treatment was the most significant. In terms of soil pH, significant effects of soil depth only occurred under forested treatments, and the effect on the 40-year-old apricot treatment was the most significant (Table 4).

**Table 4.** Effects of soil depth on soil bulk density ($\rho_s$) and pH in different treatments.

| Variables / Treatments | $\rho_s$ F | $\rho_s$ p | $\rho_s$ df | Soil pH F | Soil pH p | Soil pH df |
|---|---|---|---|---|---|---|
| Apricot (16 years) | 16.996 | 0.000 | 5 | 2.370 | 0.038 | 5 |
| Apricot (40 years) | 3.651 | 0.003 | 5 | 4.949 | 0.000 | 5 |
| Poplar (40 years) | 1.528 | 0.179 | 5 | 3.331 | 0.006 | 5 |
| Abandoned treatment | 4.214 | 0.000 | 5 | 1.138 | 0.339 | 5 |
| Cultivated treatment | 37.439 | 0.000 | 5 | 0.813 | 0.540 | 5 |

**Notes:** F = Fisher test (joint hypotheses test); *p* = probability value; df = degree of freedom.

## 4. Discussion

*4.1. The Reasons for Vertical Distribution of $\rho_s$ and pH in Different Treatments*

In cultivated treatment, $\rho_s^{0\text{-}0.1\text{ m}}$ and $\rho_s^{0.1\text{-}0.2\text{ m}}$ were low, due to frequent tillage [36] and long-term inputs of fertilizer [37]. The high $\rho_s^{0.2\text{-}0.4\text{ m}}$ can be explained by the formation of a plough-pan layer [38], in which the nutrient status was extremely poor. It is believed that the practice of organic crop rotation can be helpful in reducing plough pan reformation [39]. The increasing trend in $\rho_s$ in the upper three soil layers was the result of sustained pressure from the upper layers [40], and the lower values below 0.4 m were attributable to the declining effects of compaction from machines on the surface [41]. Our results for the cultivated treatment supported those of many studies in other regions of the world, such as Suuster et al. [17] in Estonia, and Podder et al. [39] in the Brahmaputra Floodplain, and suggest that improving soil quality in cultivated lands is a worldwide problem. The $\rho_s$ under the abandoned treatment was lower than that of the cultivated treatment in the same soil layer, and the trend in $\rho_s$ for the abandoned treatment was like that of the cultivated treatment. This suggested that although stopping farming can indeed improve soil quality, it is difficult to influence the trend of $\rho_s$ with soil depth.

In the poplar treatment and apricot treatments, $\rho_s$ in the deeper three soil layers was lower than in the upper three soil layers, a result that is contrary to the study of Zhou et al. [42], who found the lowest $\rho_s$ values to occur in the upper soil layers, due to the distribution of roots that can loosen soil. The contrasting results of the present study can be explained by the effects of compaction decreasing with soil depth. In the current study, the overall vertical distribution of $\rho_s$ was similar to that noted by Zhang et al. [43] in the North China Plain, but significantly different from that encountered by Bronick and Lal [40], who believe that due to the overburden pressure of the upper layers, larger $\rho_s$ values appear in the lower layers of the soil profile. These different results show that the relationship between $\rho_s$ and soil depth is not constant, suggesting a need for further research.

The variations of pH and $\rho_s$ with soil depth were similar. The current study suggested that this may attributable to soil pH being significantly positively correlated with $\rho_s$ (r = 0.178, $p < 0.01$). Under the apricot treatment (16 and 40 years), the soil pH of the upper layers was slightly lower than that in the deeper layers (Table 3). This phenomenon was likely related to the fine root distribution of apricot trees. Fine roots are important for the absorption of water and nutrients, and they are mainly concentrated within the 0–0.50 m soil layer [44]. According to Hong et al. [45], afforestation tends to neutralize soil pH through a mechanism of root $HCO_3^-$ secretion that occurs when plants are in acid soils, leading to a higher rhizospheric pH. Conversely, in alkaline soils, plants take up more cations than anions, releasing $H^+$ from their roots which reduces the rhizospheric pH and helps to maintain the charge balance. Therefore, in the alkaline soil of our study area [22], rhizospheric pH was slightly lower than soil pH at other depths. Similarly, under the abandoned treatment, herbaceous roots were shallower, resulting in a lower pH in the 0–0.1 m soil layer. Under the poplar treatment, pH in the 0–0.2 m soil layer was low relative to that of the deeper layers, due to litterfall decomposition (Figure 2B), which contributed organic acids to the topsoil, thereby lowering the pH [3]. The organic acids tend to break down before they can acidify the deeper layers, leading to the relatively higher soil pH in the deeper layers [38]. Poplar fine roots being mainly distributed within the 0–0.3 m soil depth [46] also contributed to the lower soil pH in the topsoil. Under the cultivated treatment, soil pH was relatively stable with depth, possibly due to a lack of fine plant roots and litterfall. Moreover, comparing our results with previous studies, we found that the relationship between soil pH and soil depth varies in different regions and under different surface vegetations. For example, Qin et al. [29] found that soil pH showed an increasing trend with depth in the Heihe River Basin, China, whereas Zhang et al. [47], investigating the vertical distribution of soil pH on a farm in Quebec, Canada, found an increasing trend of mean soil pH values with depth. In contrast, in the current study, soil pH first increased and then decreased with depth. The reason for the difference may depend on specific regional climates, soil texture and human activities [23].

### 4.2. Effects of the Stand Type on $\rho_s$ and pH

The effects of specific afforestation species on soil moisture, soil organic carbon storage, nitrogen, and phosphorus have been widely researched (e.g., [15,48–50]). However, studies of their influence on $\rho_s$ and soil pH are relatively scarce. Measurement of $\rho_s$ and pH under afforestation with different stand types was helpful in determining which stand type(s) might be best suited to environmental protection and most conducive to alleviating soil degradation in arid areas with a high susceptibility to desertification [2].

The $\rho_s$, as an important physical parameter tied to soil nutrient storage, water transportation, and gas penetration [20], was affected by several factors: AGB, compaction and the presence of roots [40,51–53]. In this study, $\rho_s$ under the 40-year-old apricot treatment (1.22 Mg m$^{-3}$) was significantly higher than under the poplar treatment of the same age (1.16 Mg m$^{-3}$), as shown in Figure 3. This may be attributed to a difference in their root systems [5]. Under the poplar treatment, SD was high (Table 2, Figure 2B), contributing to a denser root system that is able to efficiently penetrate the soil substrate, thus increasing soil porosity and reducing $\rho_s$. However, trees of the 40-year-old apricot treatment (a.k.a. "small old trees" [14]) were sparse and small (Table 2, Figure 2A$_1$), and root penetration was not as extensive as that of the poplar treatment, resulting in a higher $\rho_s$. Compaction from frequent human activities such as grazing and apricot collection (apricots are a source of income for local residents [15]) also contributed to a higher $\rho_s$ under the apricot treatments.

When comparing the $\rho_s$ of these two stand types with the cultivated and abandoned treatments, we found the $\rho_s$ of the latter to be significantly higher than those of the poplar treatment, but not significantly different from the 40-year-old apricot treatment (Figure 3). High $\rho_s$ under the cultivated treatment was attributed to the high-frequency of trampling occasioned by human activities [11,51]. Especially during the harvest season, an escalated use of agricultural machinery as well as regular human activities contributed to even greater soil compaction, further increasing $\rho_s$ [51,54]. Under the abandoned treatment, the higher $\rho_s$ may be attributable to a lack of extensive strong roots [6,52]. It was clear that the 40-year-old apricot treatment was not well suited to improving $\rho_s$, as its $\rho_s$ was not significantly lower than that measured under either the cultivated or abandoned treatment.

Soil pH in the 40-year-old apricot treatment (8.54) was significantly higher than that in the 40-year-old poplar treatment (8.52), a result tied to litterfall abundance and the acquisition of water by plants. When plants absorb groundwater, alkaline salt in the water is drawn towards the surface soil, raising soil pH [55,56]. A previous study in the same area [15] found that apricot trees consumed more water than poplars. Therefore, soil salinization under the 40-year-old apricot treatment was more serious and led to a higher soil pH. Evaporation is also an important factor in moving salts from groundwater into the topsoil [29]. During the sampling period (July to September), temperatures in the area were high, evaporation, and evapotranspiration from trees was significant, and groundwater uptake was enhanced [15], resulting in a generally higher soil salinization and soil pH. However, the crown density under the poplar treatment was significantly greater than under either of the apricot treatments (Figure 2), reducing the evaporation of soil moisture, and causing a lower soil pH in the poplar treatment. More importantly, the surface of the poplar treatment was covered with a layer of litter (Figure 2B), which decomposed to produce organic acids, contributing to a lower soil pH [57].

Compared with the cultivated treatment and abandoned treatment, the 40-year-old apricot treatment and poplar treatment generated a higher pH (Figure 3). The lower soil pH under the cultivated treatment may be attributable to the use of fertilizer. Long-term application of fertilizers—such as urea, nitrogen, and superphosphate fertilizers—would not only acidify the soil but also reduce its acid-base buffering capacity [58]. Relatively low groundwater consumption was also one of the reasons for the low pH. Crops are planted once a year in this area, so there are approximately three or four months of fallow time annually. Ground water consumption under the cultivated treatment was less than that under other treatments, and resulted in a lower soil pH. Under the abandoned treatment, although AGB was high (Table 2), water consumption by herbs was not comparable to that of trees, and the pH was relatively low, especially compared to the 40-year-old

apricot treatment. Therefore, from a soil pH perspective, neither tree species of the same age improved soil quality, with the mature, 40-year-old apricots performing the most poorly.

### 4.3. Effects of the Stand Age on $\rho_s$ and pH

The influence of stand age on $\rho_s$ has been widely studied. Previous studies have generally shown a decrease in $\rho_s$ with an increase in stand age [24,59]. However, the current study found that the mean $\rho_s$ under the 16-year-old apricot treatment was significantly lower than that under the 40-year-old treatment (Figure 3). This was likely related to: (1) the reduction of AGB and SD with increasing apricot stand age (Table 2) [52,53]; (2) greater compaction under the 40-year-old treatment [54]; and (3) root growth under the 16-year-old apricot treatment loosening the soils, whereas, after 40 years, apricot root growth had slowed or stopped, reducing their loosening effect on the soil. Soil pH also varied significantly with stand age. Under the 40-year-old apricot treatment, mean pH was significantly higher than that of the 16-year-old apricot treatment (Figure 3). The authors suggest that this may be due to the 40-year-old apricot trees' greater water requirement, which is a result of: (1) trunk thickness—the trunk of a 40-year apricot tree is thicker than that of a 16-year apricot tree [15], and (2) more AGB under the 40-year-old apricot treatment (Table 2). Jian et al. [60] found that evaporation and transpiration rates of older trees were greater, resulting in a higher soil pH through the mechanisms explained above. These results suggest that the apricot can improve soil properties at the early stage of their establishment, but later contribute to a harder soil with higher pH levels.

### 4.4. The Reasons for Effects of Soil Depth on $\rho_s$ and pH in Different Treatments

Except for the poplar treatment, the effects of soil depth on $\rho_s$ was significant. In the apricot treatment, human disturbance is the main cause of the significant effect of soil depth on $\rho_s$. As mentioned above, apricots are the main income source for local residents; accordingly, compaction from their activities at the surface resulted in different pressures being applied to the top and sub soils, leading to a significant effect of soil depth on $\rho_s$. Moreover, grazing in the apricot stands was also one of the causes of this phenomenon [50]. Under the cultivated treatment, compaction and plowing are the primary reasons for why soil depth has a significant influence on $\rho_s$. Under the abandoned treatment, although less human disturbance occurred, residual human disturbance from its prior existence as cultivated land meant that the effect of soil depth was still significant. In the study area, because poplars were forbidden to be cut down and there was less understory vegetation, there was less human and grazing disturbance; therefore, the effects of soil depth were not as significant as under other treatments.

Under forested treatments, the effect of soil depth on pH was significant, while the same effects did not occur under non-forested treatments. This may be related to the differences in plant populations: under forested treatments, due to the distribution of the roots, the soil pH at different soil depths differed significantly [44,45]; however, under the abandoned treatment, the effect of roots may be not as significant as under forested treatments. In cultivated treatments, our result was similar to the result of Anderson et al. [61], who found that soil pH is not stratified in conventionally tilled fields. A possible reason for this is that ploughing made soil pH more uniform.

### 4.5. Interaction Effects Between Treatment and Soil Depth on $\rho_s$ and pH

Interactions between plants, soils, and microorganisms regulate the function of terrestrial ecosystems, and biotic and abiotic interactions strongly impact ecological processes [62]. Many studies have explored the effects of these interactions on variables. For example, Strobel et al. [63] found that the release rate of copper from soil was correlated with the interaction between soil pH and concentrations of dissolved organic matter in afforestation lands; Hu et al. [64] found that leaf nitrogen and C: N (carbon: nitrogen) were significantly affected by the interaction between stand age and desertification intensity; Qin et al. [29] found that soil organic carbon was not affected by the interaction between soil type and slope aspect, while Dearborn and Danby [65] found that the species richness

was not affected by the interaction between altitude and the slope aspect. However, research on the interaction effects between treatments and soil depth on $\rho_s$ and pH are largely absent from the literature. To the best of the authors' knowledge, only one study, by Menyailo et al. [66], was similar to the present study. In their work, no interaction existed between the effects of stand type and soil depth on net N mineralization.

In the current study, we used treatment and soil depth as fixed factors. The current study showed significant effects on $\rho_s$ arising from interactions between treatment and soil depth, whereas soil pH was not affected by these interactions (Table 5). Given the dearth of research on these specific interactions, further studies are required to support the results of the current investigation.

**Table 5.** Interaction effects between soil depth and treatment on soil bulk density ($\rho_s$) and pH.

| Factors / Variables | Soil depth | | | Treatment | | | Soil depth $\times$ Treatment | | |
|---|---|---|---|---|---|---|---|---|---|
| | F | p | df | F | p | df | F | p | df |
| $\rho_s$ | 19.804 | 0.000 | 5 | 108.043 | 0.000 | 4 | 8.360 | 0.000 | 20 |
| Soil pH | 9.732 | 0.000 | 5 | 35.992 | 0.000 | 4 | 1.382 | 0.119 | 20 |

Notes: F = Fisher test (joint hypotheses test); *p* = probability value; df = degree of freedom.

In summary, the $\rho_s$ and pH values of the poplar treatment were lower than those of the 40-year-old apricot treatment, while the $\rho_s$ and pH of the 40-year-old apricot treatment were higher than those of the 16-year-old apricot treatment. This suggests that the stand type and stand age can both affect soil properties, and that the poplar and the 16-year-old apricot treatments contributed the most to improving soil quality in the study area. Cao et al. [15] reported that a 40-year-old poplar stand consumed less water than a 16-year-old apricot stand situated in the same area. As water is a limited resource in the arid and semiarid region under study [67,68], poplars may be the species best-suited for afforestation in this area. However, afforestation efforts focus not only on water conservation, but also on carbon sequestration. This suggests a need for a more measured, comprehensive, and holistic approach to the assessment of afforestation benefits [69]. In addition, as treatment and soil depth interacted to affect $\rho_s$ and pH, these factors must not be neglected in future assessments of the integrated benefits of afforestation.

## 5. Conclusions

In this study, the mean soil bulk density in the 0–1.0 m layer under different forested treatments, excluding that of the 40-year-old apricot treatment, was lower than that of either the abandoned or the cultivated treatments. The mean soil pH in the 0–1.0 m layer was higher under forested treatments compared to the cultivated treatment. In comparing $\rho_s$ and pH across different stand types of the same age, the poplar treatment showed a superior capacity to enhance soil properties. The soil bulk density and pH showed strong increasing trends with apricot stand age. Based on these results, the authors concluded that both the stand type and stand age are important factors in lowering soil bulk density and pH in the study region. As an older apricot stand can contribute to a decline in soil quality, the authors recommend that the regional afforestation policy with apricot as the main tree species should be prudently implemented. In addition, treatment and soil depth had interactive effects on soil bulk density, warranting the inclusion of these factors in any soil bulk density investigation in the area. These same factors were found to have no interactive effect on pH.

As the study was conducted in a relatively small area on the Loess Plateau, further research across a range of diverse districts is needed to produce more extensive results and insights that can be applied to the resolution of soil problems on the vast Loess Plateau.

**Author Contributions:** Conceptualization, X.Z. and J.C.; Methodology, J.C.; Validation, J.C.; Formal Analysis, X.Z. and X.X.; Investigation, X.Z., J.C. and X.X.; Data Curation, X.Z. and X.X.; Writing-Original Draft Preparation, X.Z.; Writing-Review & Editing, J.F.A., R.C.D., G.Z. and J.C.; Visualization, X.Z.; Funding Acquisition, J.C.

**Funding:** This research was funded by the National Natural Science Foundation of China (41461109), the Major Program of the Natural Science Foundation of Gansu province, China (18JR4RA002), and the Key Laboratory of Ecohydrology of Inland River Basin, Chinese Academy of Science (KLERB-ZS-16-01).

**Conflicts of Interest:** The authors declare no conflict of interest.

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
