# Peer review of "Effects of Afforestation on Soil Bulk Density and pH in the Loess Plateau, China"

_water, doi:10.3390/w10121710_

Round 1

Reviewer 1 Report

The article can be published after some editing.

1.       It is better to replace the abbreviation ρs on density in the conclusion and summary

Need a map of China with the location of objects

2.       As can be seen from the photo in Figure 2D, the arable land is very stony, how did the authors determine the density there? And what is the rockiness on other sites?

3.       The plots locate on large slope,  need to specify where the plots were located: at the beginning of the slope, in the middle or below ?.

4.       In Table 1  does not indicate for which layer the density and pH are determined, and what mean the numbers 1, 2, 3 and df

5.       Table 2. need  give data as 1029,   not 1029,63

6.       Table 3. At the Apricot site (16 years), the density decreases at a depth starting from 0.4 m and especially in 0.8-1 m. And in other areas at a depth of 0.8-1 m, the density is greatly reduced, this layer should be excluded from the calculations and only a depth of 0-0.8 m should be considered.

7.       The differences in density between all sites are very small and do not exceed the limits of the permissible soil density for growing plants and the danger of increasing erosion.

8.       6. Density depends on the particle size distribution of the soil, it would be desirable to provide these data

9.       Lines 88, 89 The content of sludge and sand is about 60%. Specify data separately for sludge and separately for sand.

10. Some  references recommended

The influence of spruce on acidity and nutrient content in soils of Northern Taiga dwarf shrub–green moss spruce forests

// Eurasian Soil Science.  Vol 49, Is.11, pp 1276–1287. DOI: 10.1134/S1064229316110077

Basal respiration and composition of microbial biomass in virgin and agroforest-reclaimed semidesert soils of the Northern Caspian region // Eurasian Soil Science,  Vol. 48, No. 8, pp. 852–861. DOI: 10.1134/S1064229315080049

Author Response

Comment

The article can be published after some editing.

1.       It is better to replace the abbreviation ρs on density in the conclusion and summary

Need a map of China with the location of objects.

Response

Thank you for your suggestion, and we have replaced ρ_s with soil bulk density in conclusion section. In Figure 1, the map of China was also inserted.

Comment

2.       As can be seen from the photo in Figure 2D, the arable land is very stony, how did the authors determine the density there? And what is the rockiness on other sites?

Response

The cultivated lands as well as other land types, are not stony, and those which look like stones in figure 2D were clods.

Comment

3.       The plots locate on large slope, need to specify where the plots were located: at the beginning of the slope, in the middle or below ?

Response

Plots were located in the middle slope, and we have added these information in the text (Page 3, Line 127)

Comment

4.       In Table 1  does not indicate for which layer the density and pH are determined, and what mean the numbers 1, 2, 3 and df.

Response

Soil bulk density and pH in the 0-1.0 m soil layer are determined. The numbers 1, 2, 3, and df represent block 1, block 2 and block 3, and degree of freedom, respectively. We have added these information in tables, including the representation of ‘df’ in Table 3, 4, 5.

Comment

5.       Table 2. need  give data as 1029,   not 1029,63

Response

Thank you for your correction, and we have changed.

Comment

6.       Table 3. At the Apricot site (16 years), the density decreases at a depth starting from 0.4 m and especially in 0.8-1 m. And in other areas at a depth of 0.8-1 m, the density is greatly reduced, this layer should be excluded from the calculations and only a depth of 0-0.8 m was considered.

Response

As many studies, the soil depth in this paper was 1 m. In Table 3. If the 0.8-1.0 m soil layer was excluded from the calculations and only a depth of 0-0.8 m was considered when studying soil bulk density, it will firstly cause inconsistencies of the data across all sections of the text. Secondly, it will make readers confused: Why does the author omit the data of soil bulk density of the 0.8-1.0 m soil layer? Is the data of this layer not processed? Thereby, excluding this layer will affect the reliability of the results. As a result, we did not exclude 0.8-1.0 m soil layer.

Comment

7.       The differences in density between all sites are very small and do not exceed the limits of the permissible soil density for growing plants and the danger of increasing erosion.

Response

Surely, the differences in density between all sites are very small, but whether they can affect plants growth and soil erosion or not, is needed further study in future.

Comment

8.        Density depends on the particle size distribution of the soil, it would be desirable to provide these data

Response

Thank you for your suggestion. We have added these information in the text: The soil particles are dominated by silt (above 60 %), and clay and sand accounts for 5%, and about 30%, respectively. (Page 3 Line 88-90)

Comment

9.       Lines 88, 89 The content of sludge and sand is about 60%. Specify data separately for sludge and separately for sand.

Response

In order to avoid confusion, we deleted this sentence, and added the particle size distribution information as shown above (response to the comment 8). 

Comment

10. Some  references recommended 

The influence of spruce on acidity and nutrient content in soils of Northern Taiga dwarf shrub–green moss spruce forests

// Eurasian Soil Science.  Vol 49, Is.11, pp 1276–1287. DOI: 10.1134/S1064229316110077

Basal respiration and composition of microbial biomass in virgin and agroforest-reclaimed semidesert soils of the Northern Caspian region // Eurasian Soil Science,  Vol. 48, No. 8, pp. 852–861. DOI: 10.1134/S1064229315080049

Response

Thank you for your kindness. However, due to the differences in species and research areas, the references are not valuable for this paper citing.

Reviewer 2 Report

Dear authors,

For a better understanding of the results, I suggest to the authors to improve the section "Study Area" regarding the soil description in relationship with geological deposits (loess) and the landforms in which the profiles are located. In the figure 2 it is clear that some sites are located on slopes, and the soil properties are linked with slope processes. For the figure 2, I suggest to provide a small vignette for framing the study area in China territory. For the figure 3 I suggest to find a more expressive way (change the type of graphic).

Best regards

Author Response

Comment

For a better understanding of the results, I suggest to the authors to improve the section "Study Area" regarding the soil description in relationship with geological deposits (loess) and the landforms in which the profiles are located.

Response

Thank you for your suggestion, we have added these information in the text: The region is characterized by complex tectonic structures, most of which are based on metamorphic rocks and granites. The Quaternary deposits (loess) are widely distributed as landform types of low-hill, loess-hill and loess hill-ridge, accounting for more than 95% of the total area. (Page 3, Line 85-88)

Comment

In the figure 2 it is clear that some sites are located on slopes, and the soil properties are linked with slope processes.

Response

In order to avoid the effects of different slope positions on soil properties, the plots located in the middle slope were selected during our field investigation. We have added these information in the text (Page 3, Line 127)

Comment

For the figure 1, I suggest to provide a small vignette for framing the study area in China territory.

Response

Thank you for your suggestion, and this piece of advice is also put forward by reviewer 1. We have provided a more detailed map (Figure 1) for framing the study area in China territory.

Comment

For the figure 3 I suggest to find a more expressive way (change the type of graphic).

Response

In order to make a more expressive way, we have integrated two Figures into one, and substituted bar graph with scatter graph.

Reviewer 3 Report

This is a well written manuscript dealing with impacts of land management on soil quality in an area of China that has experienced land degradation over centuries.  Specific comments are:

1.      The differences in bulk density and pH, though significant, and small throughout the soil profile.  However, most of these soils have been drastically disturbed by humans in the past when these lands were shaped by forming terraces for agricultural production.  This may be why the differences in both bulk density and pH are so small throughout the 1 m profile.  The authors should highlight this in their study area description as well as give an estimate of time since the land was terraced.  This would add valuable context to the manuscript.

2.      Page 10, lines 338-339:  “…not comp cultivated…”?  Something is missing here.  Please check.

This manuscript should be accepted for publication after minor revision

Author Response

Comment

This is a well written manuscript dealing with impacts of land management on soil quality in an area of China that has experienced land degradation over centuries.  Specific comments are:

1.      The differences in bulk density and pH, though significant, and small throughout the soil profile.  However, most of these soils have been drastically disturbed by humans in the past when these lands were shaped by forming terraces for agricultural production.  This may be why the differences in both bulk density and pH are so small throughout the 1 m profile.  The authors should highlight this in their study area description as well as give an estimate of time since the land was terraced.  This would add valuable context to the manuscript.

Response

Yes, the soil in this area have been drastically disturbed by humans when these lands were shaped by forming terraces in the 1970s. We have added these information in the text. (Page 3, Line 91-94)

Comment

2.      Page 10, lines 338-339:  “…not comp cultivated…”?  Something is missing here.  Please check. 

Response

Thank you for your correction, we have corrected it.

This manuscript should be accepted for publication after minor revision